# Robust Pedestrian Detection Based on Multi-Spectral Image Fusion and Convolutional Neural Networks

**Xu Chen, Lei Liu \*** and **Xin Tan**

School of Electronic Engineering and Optoelectronic Technology, Nanjing University of Science and Technology, Nanjing 210094, China; chenxu321194@163.com (X.C.); fddbnsbstx@163.com (X.T.)
\* Correspondence: liulei442@njust.edu.cn; Tel.: +86-25-84314969

**Abstract:** Nowadays, pedestrian detection is widely used in fields such as driving assistance and video surveillance with the progression of technology. However, although the research of single-modal visible pedestrian detection has been very mature, it is still not enough to meet the demand of pedestrian detection at all times. Thus, a multi-spectral pedestrian detection method via image fusion and convolutional neural networks is proposed in this paper. The infrared intensity distribution and visible appearance features are retained with a total variation model based on local structure transfer, and pedestrian detection is realized with the multi-spectral fusion results and the target detection network YOLOv3. The detection performance of the proposed method is evaluated and compared with the detection methods based on the other four pixel-level fusion algorithms and two fusion network architectures. The results attest that our method has superior detection performance, which can detect pedestrian targets robustly even in the case of harsh illumination conditions and cluttered backgrounds.

**Keywords:** pedestrian detection; multi-spectral; image fusion; convolutional neural network

## 1. Introduction

As an important task in target detection, pedestrian detection is widely used in traffic safety, video surveillance, human–computer interaction, and other fields [1–3]. So far, there have been many pedestrian detection methods, most of which are based on a visible image. With the continuous progress of deep learning, target detection algorithms such as Faster Region-Based Convolutional Neural Networks (Fast-RCNNs) [4], Single-Shot Detection (SSD) [5], and You Only Look Once (YOLO) [6–8] have been proposed one after another, and the technology of pedestrian detection has achieved unprecedented development. However, these single-modal visual methods do not perform well in complex scenes, such as in poor lighting conditions and chaotic backgrounds. How to enhance the robustness of pedestrian detection in complex scenes is still a huge challenge.

Infrared images are obtained by capturing the thermal radiation emitted by objects, which are less influenced by external conditions such as illumination, making robust pedestrian detection in complex scenes possible. However, infrared images lack details and reflect limited texture information. Detection based on visible images usually achieves better performance under good illumination conditions due to abundant appearance details. Therefore, how to effectively combine visible information and infrared information is a key issue for multi-spectral pedestrian detection.

Over the years, a variety of methods for multi-spectral image fusion has been proposed [9,10]. These methods mainly include multi-scale transformation (MDT) [11–13], sparse representation [14,15], subspace [16,17], saliency [18,19], and deep networks [20–22]. In recent years, deep network-based fusion methods have become a popular topic of research, but these methods are usually based on complex computational models and require a large amount of multi-spectral fusion data. These limitations make the network-based fusion methods encounter many difficulties when combined with other tasks or applications.

In contrast, MST-based methods have been more extensively studied and widely used in different fields.

Multifarious MST-based fusion methods have been proposed, such as a wavelet transform (WT) [23], curvelet transform (CVT) [24,25], local edge-preserving filter (LEP) [26], and weighted least squares filter (WLS) [27,28]. Additionally, numerous research has attested that MST fusion methods are consistent with human visual perception. However, these methods often ignore the differences between multispectral images and extract similar salient features without discrimination, which sometimes makes the targets not prominent enough in fused images. To address this issue, a fusion method based on the total variation (TV) was proposed by Ma et al. [29], which adopted different feature representations from the source image pairs. The fusion images were obtained by combining the intensity features of the infrared image and the gradient features of the visible image. Zhang et al. [30] achieved image fusion by extracting infrared features and preserving visual information, which not only showed the infrared thermal objects but also retained a good deal of the visual details. Kong et al. [31] used the idea of guided filtering to transmit the structure information of visible images to infrared images and obtained fusion images with obvious targets and realistic textures.

With the development of target detection networks, multi-spectral pedestrian detection methods via convolutional neural networks have gradually attracted attention [32–36]. Wagner et al. [37] first proposed a multi-spectral pedestrian detection model based on early fusion and late fusion architectures on the basis of an RCNN. Liu et al. [38] studied and analyzed the impacts of the fusion at different stages of the networks on detection performance based on a Fast-RCNN, including early fusion, halfway fusion, late fusion, and confidence fusion architectures. Cao et al. [39] proposed an unsupervised learning method for a DNN-based pedestrian detector, realizing the unsupervised learning of multi-spectral features by an automatic labeling method. Chen et al. [40] proposed a multilayer fused deconvolutional single-shot detector based on a two-stream convolutional module and a multilayer fused deconvolutional module, improving the computational efficiency and detection accuracy of small-sized targets.

However, the above research only discusses fusion based on deep networks and lacks in-depth exploration of target detection based on pixel-level fusion. Hou et al. [41] tested the application of pixel-level image fusion methods in an SSD target detector and verified the significance of pixel-level image fusion in multi-spectral pedestrian detection. However, the pixel-level image fusion methods tested in their research cannot effectively utilize the characteristic information from infrared images and visible images. Therefore, a multi-spectral pedestrian detection method via pixel-level image fusion is proposed in this paper, using the intensity features in infrared images and the local structure features in visible images combined with the target detection network YOLOv3 to enhance the accuracy and robustness of pedestrian detection.

The contribution of this paper lies in the following four aspects:

(1) In light of the insufficient research on multi-spectral image fusion in the existing multi-spectral pedestrian detection methods, we further study the multi-spectral pedestrian detection methods by using pixel-level image fusion. In addition, a multi-spectral pedestrian detection method based on pixel-level image fusion and a convolutional neural network is proposed. This method makes full use of the different feature information of infrared images and visible images and combines YOLOv3 to achieve robust pedestrian detection.

(2) Aiming at the information loss caused by mutual cancellation of opposite information when infrared and visible light images are fused, a multi-spectral image fusion method via TV minimization and local structure transfer is proposed. This method effectively preserves the intensity distribution of infrared images and the local structural features of visible images. In addition, an infrared detail enhancement method is introduced to increase the detail information of the thermal target area. The fusion image can

highlight pedestrian targets and retain abundant appearance information, which is conducive to pedestrian detection.

(3)   Two fusion architectures based on YOLOv3 are designed and implemented for comparison, namely early fusion and late fusion. Multi-spectral pedestrian detection is realized by the fusion of features with different scales at different network depths in YOLOv3.

(4)   We qualitatively and quantitatively compare and evaluate the detection results of our proposed method with four pixel-level fusion methods and two fusion network architectures. The experimental results illustrate that our proposed method effectively improves the robustness and accuracy of pedestrian detection, especially under harsh visual conditions.

## 2. Proposed Method

YOLOv3 [8] is a single-stage target detection method whose backbone network uses Darknet-53 without a pooling layer or full connection layer. YOLOv3 uses structures similar to the residual network and the feature pyramid network, which can achieve good performance in terms of accuracy and rate of detection. Therefore, this paper adopted YOLOv3 as the benchmark system in our evaluation.

The flow diagram of the proposed method is presented in Figure 1. First, HSI color space transformation is performed on the RGB images [42]. Secondly, the TV minimization method based on structure transfer is adopted to fuse the I component of the converted image and the infrared image. Then, the fusion result replaces the original I component, and the HSI image is converted into an RGB image to receive the color fusion result. Finally, the color fusion result is input into the YOLOv3 network to implement multi-spectral pedestrian detection. Aside from that, this paper also designs and implements the early fusion architecture and the late fusion architecture based on YOLOv3 for comparison.

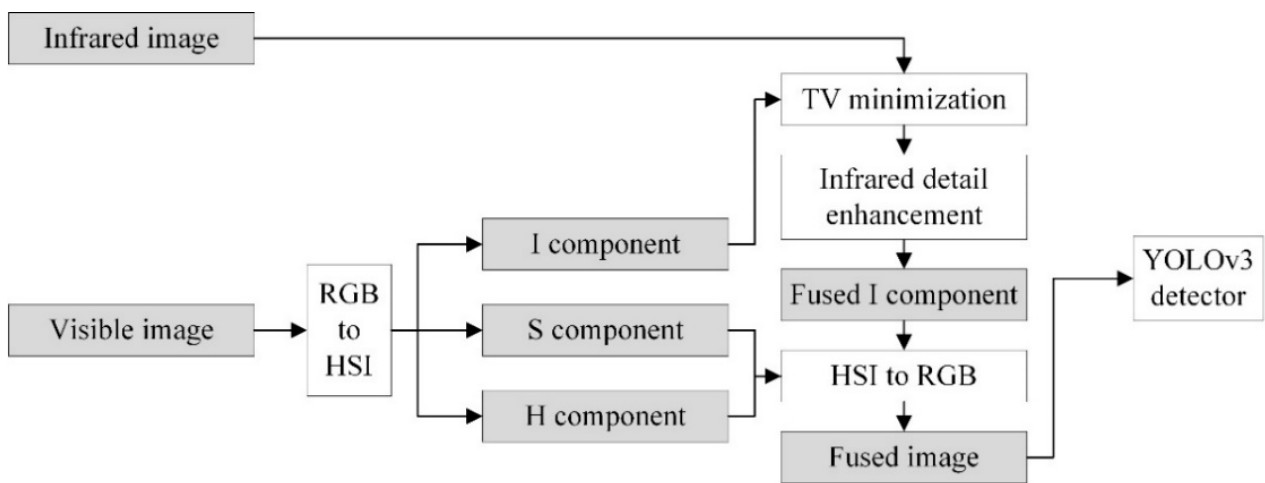

**Figure 1.** The flow diagram of the proposed method.

### 2.1. Pedestrian Detection Based on Pixel-Level Color Image Fusion

2.1.1. Color Space Transformation

The majority of the existing color image fusion methods are based on color space transformation. The most common color spaces in the field of image processing include RGB, LAB [43], Ycbcr [44,45], and HSI [46,47]. These methods usually transform the RGB visible image into another color space to obtain different image components and then fuse one of the image components with the infrared image. Among them, LAB-based methods convert the visible image from RGB to LAB and fuse the L component with the infrared image to get a new L component. Ycbcr-based methods convert the visible image from RGB to Ycbcr and fuse the Y component with the infrared image to get a new Y component. HSI-

based methods convert the visible image from RGB to HSI and fuse the I component with the infrared image to get a new I component. In this paper, we chose the most appropriate color space transformation method after a comprehensive comparison of the fusion results based on LAB, Ycbcr, and HSI in terms of target saliency, color authenticity, and detail clarity. The fusion results via different color spaces are exhibited in Figure 2.

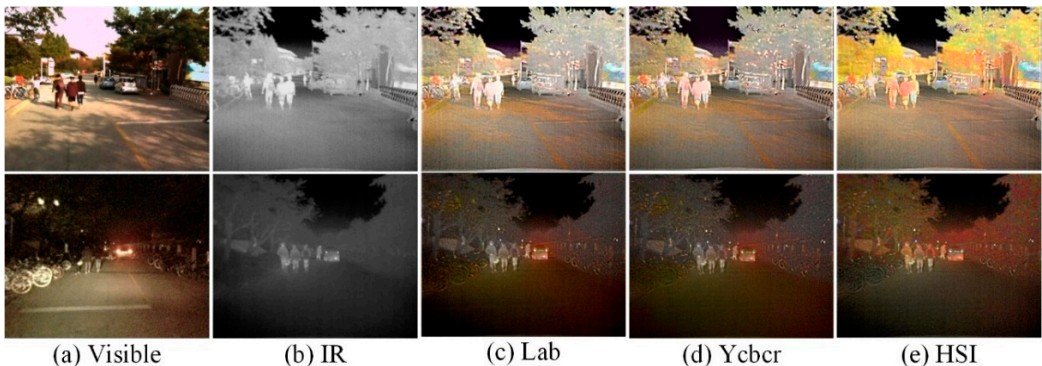

(a) Visible      (b) IR      (c) Lab      (d) Ycbcr      (e) HSI

**Figure 2.** The fusion results via different color spaces. (**a**) shows the visible images; (**b**) shows the infrared images; (**c**–**e**) respectively show the fusion results based on Lab, Ycbcr and HIS.

As can be seen from the color fusion images, the color fusion based on LAB was slightly better than that of YCbcr in color transmission, but the overall difference was not significant. The HSI-based method had the best color transfer performance, and the fusion results had more prominent features and more details in the target region. Therefore, this paper adopted the HSI color space transformation method for color image fusion.

The HSI color space uses three mutually independent features to describe colors, namely hue (*H*), saturation (*S*), and intensity (*I*), which are more consistent with human visual characteristics than the RGB color space. In this paper, a standard model method [48] was adopted to convert RGB images into the HIS color space, and the specific formulas are as follows:

$$H' = \begin{cases} \frac{\pi}{3} \times \frac{G-B}{T_{\max} - T_{\min}}, & if \ T_{\max} = R \\ \frac{\pi}{3} \times \frac{B-R}{T_{\max} - T_{\min}}, & if \ T_{\max} = G \\ \frac{\pi}{3} \times \frac{R-G}{T_{\max} - T_{\min}}, & if \ T_{\max} = B \end{cases} \tag{1}$$

$$H = \begin{cases} H', & if \ H' \geq 0 \\ H' + 2\pi, & if \ H' < 0 \end{cases} \tag{2}$$

$$S = T_{\max} - T_{\min} \tag{3}$$

$$I = (T_{\max} + T_{\min})/2 \tag{4}$$

where $T_{\max} = \max(R, G, B)$ and $T_{\min} = \min(R, G, B)$.

2.1.2. Image Fusion Based on TV Minimization and Structure Transfer

The features extracted from the same area of infrared images and visible images sometimes convey reverse information. The traditional fusion methods retain these features without distinction, which will bring about the loss of some crucial information. The fusion method using gradient transfer [29] transmits the gradient details from the visible images to the corresponding positions of infrared images, which preserves the intensity distribution in infrared images and the detail gradients in visible images concurrently while almost completely neglecting the intensity information of the visible images, resulting in the loss of the appearance details of the targets. The local structure usually plays a stronger role in detail expression, which is more consistent with the visual perception of humans [31]. Hence, we replaced the gradient features with the local structures, making the fused results

have local structures similar to those of the visible images. The fusion task can be expressed as the minimization of the following objective functions:

$$E(F_{\omega_k}) = \|F_{\omega_k} - I_{\omega_k}\|_1^1 + \lambda \left\| \frac{\nabla F_{\omega_k}}{F_{\omega_k}} - \frac{\nabla V_{\omega_k}}{V_{\omega_k}} \right\|_1^1 \tag{5}$$

where $\omega_k$ denotes the window centered on pixel $k$, $F_{\omega_k}$ and $I_{\omega_k}$ denote the fused result and the infrared image in $\omega_k$, respectively, $\overline{F_{\omega_k}}$ and $\overline{V_{\omega_k}}$ denote the mean values of $F_{\omega_k}$ and the visible image in $\omega_k$, respectively, and $\frac{\nabla BF_{\omega_k}}{BF_{\omega_k}}$ and $\frac{\nabla BV_{\omega_k}}{BV_{\omega_k}}$ represent the local structures of the fussed result and the visible image, respectively. On the right side of the formula, the first item constrains the fusion result to having a similar intensity distribution with that of the infrared image, and the second item constrains the fusion result to have a similar gradient with the visible image. $\lambda$ denotes the regularization parameter which regulates the trade-off between two items, and we set $\lambda = 2$ here.

Since the intensity distribution of the fusion result resembled that of the infrared image, we assumed that $\overline{F_{\omega_k}} \approx \overline{I_{\omega_k}}$. Let $y_{\omega_k} = BF_{\omega_k} - \frac{\overline{BI_{\omega_k}}}{\overline{BV_{\omega_k}}} \cdot BV_{\omega_k}$. Then, the optimization problem can be expressed as

$$y_{\omega_k}^* = \underset{y_{\omega_k}}{\arg\min} \left\{ \sum_{i \in \omega_k} \left\| y_{\omega_k}^i - \left( I_{\omega_k}^i - \frac{\overline{I_{\omega_k}}}{\overline{V_{\omega_k}}} \cdot V_{\omega_k}^i \right) \right\|_1^1 + \frac{\lambda}{\overline{I_{\omega_k}}} J(y_{\omega_k}) \right\} \tag{6}$$

where $J(y_{\omega_k}) = \sum_{i \in \omega_k} \|\nabla_i y_{\omega_k}\|_1^1$. The global optimal solution of the fused image in $\omega_k$ can be expressed as

$$F_{\omega_k}^* = y_{\omega_k}^* + \frac{\overline{I_{\omega_k}}}{\overline{V_{\omega_k}}} \cdot V_{\omega_k} \tag{7}$$

The final fused image can be obtained by combining the fused image of each window, namely $F$.

### 2.1.3. Infrared Detail Enhancement

Figure 3c exhibits the fusion results procured by the above method, which retained an intensity distribution similar to the infrared images and the local structure features of the visible images. Nevertheless, due to the trade-off of these two kinds of information, the details of the fusion results were not clear enough, which may have affected the subsequent target detection, especially the detection of small-scale targets. For small-scale targets, there are few appearance features in visible images which are greatly affected by illumination, while the intensity information in infrared images makes small-scale targets still significant. Therefore, the infrared detail enhancement of the initial fusion results was beneficial for improving the target detection performance.

Inspired by the authors of [47,49], the mean filter and Gaussian filter were used to extract the infrared details of different scales, which were combined with the initial fused result to get the final fused result. The formula can be expressed as follows:

$$F_E = F + \alpha_1 \cdot DF_{a(r_a)} + \alpha_2 \cdot DF_{g(r_{ga}, \sigma)} \tag{8}$$

where $F_E$ represents the final fusion image and $\alpha_1$ and $\alpha_2$ are constant coefficients. $DF_{a(r_a)}$ and $DF_{g(r_{ga}, \sigma)}$ represent the detail information extracted by the mean filter and Gaussian filter, respectively, the formulas of which can be presented as follows:

$$DF_{a(r_a)} = F - avefilter(F, r_a) \tag{9}$$

$$DF_{g(r_{ga}, \sigma)} = F - gaussian(F, r_{ga}, \sigma) \tag{10}$$

where $r_a$ and $r_{ga}$ denote the sizes of the mean filter and Gaussian filter, respectively, and $\sigma$ denotes the standard deviation. In this paper, we set $\alpha_1 = \alpha_2 = 0.5$, $r_a = 15$, $r_{ga} = 7$, and $\sigma = 3$.

Figure 3d shows the examples of the final fusion results, where we can see that the enhancement of the infrared details made the target areas of the fusion results more significant and the details clearer.

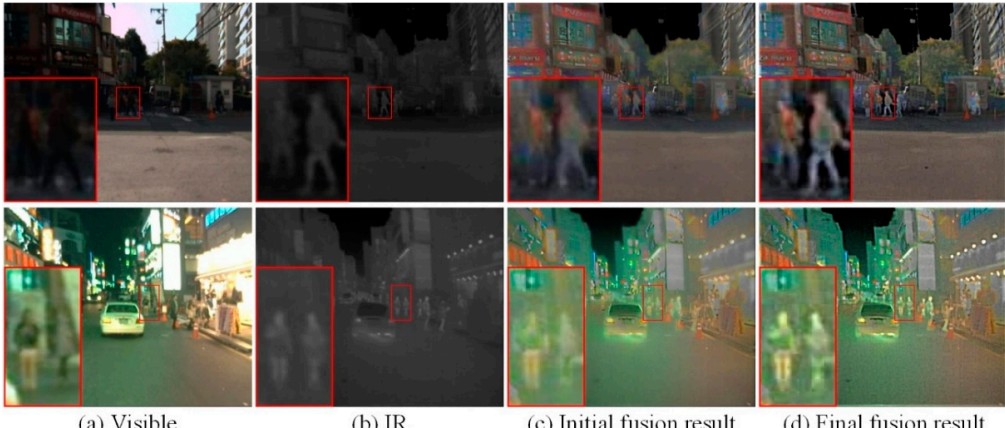

Figure 3. The fusion results of the proposed method. (a) shows the visible images; (b) shows the infrared images; (c) shows the initial fusion results without infrared detail enhancement; (d) shows the final fusion results with infrared detail enhancement.

### 2.1.4. Pedestrian Detection Based on Multi-Spectral Fusion Results and YOLOv3

The fusion images of the multi-spectral pedestrian dataset were used as the input images of YOLOv3 to train and test the model of multi-spectral pedestrian detection. For the training of the YOLOv3 detector, the weights of Darknet-53 were pretrained with the ImageNet dataset, while the weights of other convolutional layers were initialized by Kaiming initialization [50]. Then, the multi-spectral pedestrian dataset KAIST [33] was used to fine-tune the detection. The stochastic gradient descent algorithm (SGD) was used to fine-tune the model, the batch size was set to be 4, the momentum was set to be 0.9, the decay was set to be 0.0005, and the learning rate was set to be 0.001. Aside from that, the convolutional layers of Darknet-53 were frozen during the fine-tuning training for the first 20 epochs. Then, the freezing was canceled, and all layers participated in the training until the fine-tuning was completed.

### 2.2. Pedestrian Detection Based on Fusion Architectures

For better verification of the performance of our proposed method, we not only adopted some pixel-level fusion methods for comparison but also implemented two fusion architectures based on YOLOv3, namely early fusion and late fusion. The early fusion architecture is shown in Figure 4, where visible images and infrared images were cascaded as an input with four channels for YOLOv3.The network structure was the same as the original YOLOv3, with the exception of the first convolutional layer.

For the training of this model, the weights of the above-mentioned pre-trained model were first loaded to all layers of the backbone network except the first convolutional layer. Then, the Kaiming initialization method was adopted to initialize the weights of other convolutional layers, and the KAIST dataset was used to fine-tune the model. The other settings were the same as those in the above method.

Figure 5 exhibits the late fusion architecture based on YOLOv3. In this architecture, visible images and infrared images were the input to Darknet-53 to extract the respective features, and the feature maps of different scales were fused in different depths of the networks. Then, the fusion images of different scales were classified respectively to realize pedestrian detection. For feature map fusion, the authors of [51] explored three fusion strategies, namely cat, max, and sum. The study indicated that the most commonly used cat

fusion strategy performed the worst, and the sum strategy performed the best. Therefore, this paper used the sum strategy to fuse the feature maps.

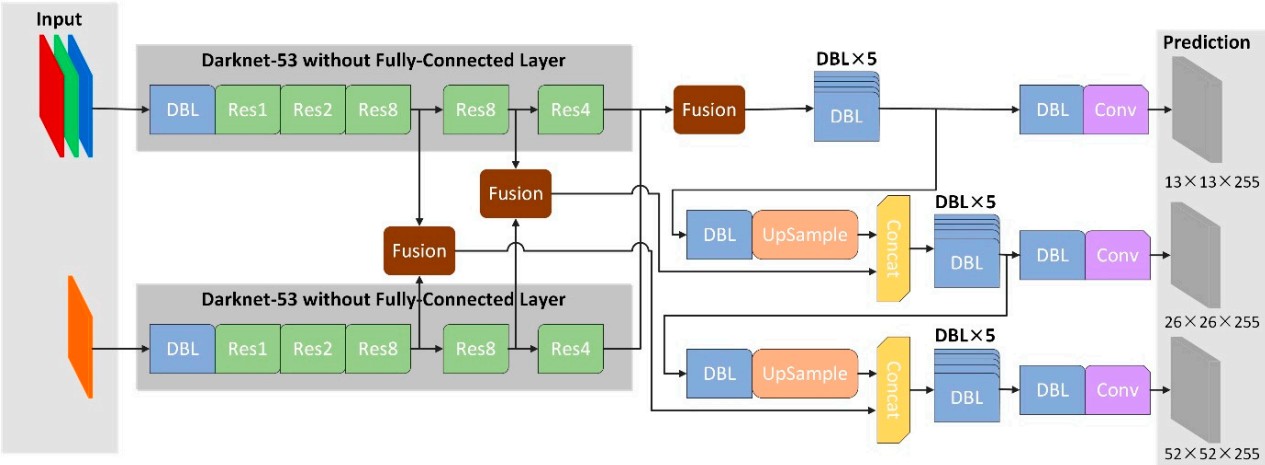

**Figure 4.** Early fusion architecture based on YOLOv3.

**Figure 5.** Late fusion architecture based on YOLOv3.

Inspired by the authors of [41], the pre-trained weights model was assigned to the subnetwork that extracted the features of visible images. The weights of the subnetwork that extracted the features of infrared images and other layers were initialized by using the Kaiming initialization method. Then, the KAIST dataset was used for fine-tuning. Compared with the original YOLOv3, the network structure of the late fusion architecture has changed a lot and become more complex, so it is necessary to spend more time on training. The other settings were the same as those in the above method.

## 3. Experiments and Analysis

### 3.1. Datasets and Settings

The publicly available multispectral pedestrian detection dataset KAIST [33] consists of 95,328 pairs of aligned visible images and infrared images with 103,128 pedestrian labels. This dataset captures a variety of routine traffic scenes including campus, street, and downtown scenarios during the day and night. This paper adopts the cleaned dataset to train and evaluate the detector, which includes 7601 train set pictures and 2252 test set pictures [38]. In order to simplify the implementation, processing such as data augmentation is not used for the dataset.

Since the focus of this work was to evaluate the influence of different fusion methods on the detection results, the settings of the network remained the same as those of the

original YOLOv3. The experiments were executed on a desktop with a GTX 1080ti GPU, 11 GB memory, and the batch size set to be 4.

### 3.2. Comparison of the Fused Results via Different Fusion Methods

In order to prove the performance of the proposed fusion method, the fusion results of this method were qualitatively and quantitively compared with those of CVT [25], IFEVIP [30], VSMWLS [28], and STF [31]. The qualitative fusion results with different methods are shown in Figure 6, in which the first three rows show the daytime images, and the last three rows show the nighttime images.

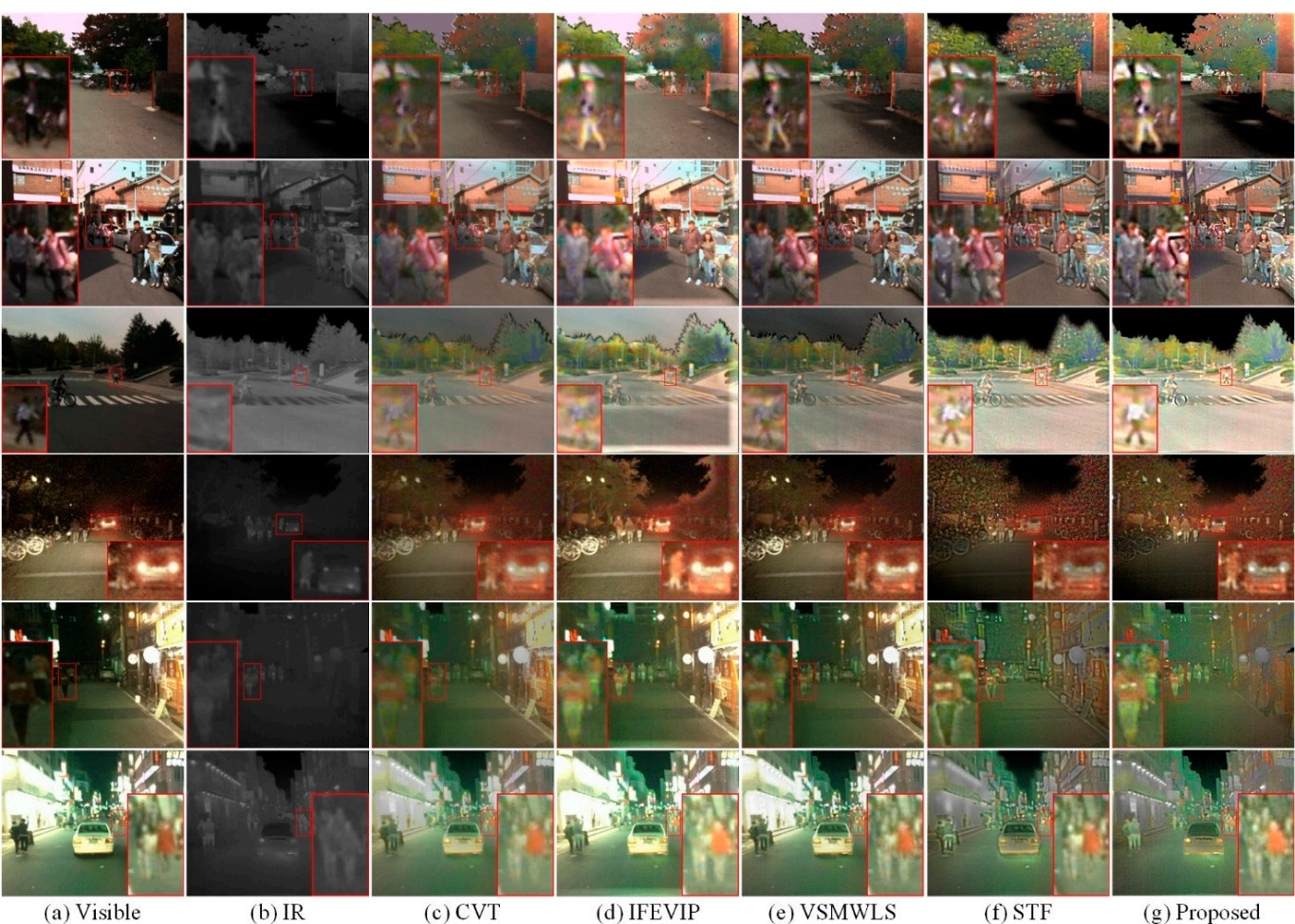

(a) Visible　　(b) IR　　(c) CVT　　(d) IFEVIP　　(e) VSMWLS　　(f) STF　　(g) Proposed

**Figure 6.** Qualitive comparison of fusion results with different methods. (**a**) shows the visible images; (**b**) shows the infrared images; (**c–g**) respectively show the fusion results based on CVT, IFEVIP, VSMWLS, STF and the proposed method.

It can be seen from the figure that, compared with the other methods, the targets of the CVT fusion results were not significant enough, and the details were blurred, which is not conducive to target detection. IFEVIP retained the features of the infrared thermal target and abundant background visual information but could not highlight the pedestrian targets when the visible background was cluttered, such as the fusion result in the sixth row. VSMWLS balanced the infrared information and the visible information well, which is consistent with human vision. However, the targets were still not prominent enough under poor illumination conditions, and the detection was susceptible to interference from background information. STF retained a large amount of visible information, but the targets were not prominent and indistinct. The proposed method retained the intensity distribution similar to that in infrared images and the local structural features in visible images, which

made the thermal targets maintain a high contrast with the background, and the target areas had abundant appearance information.

That aside, it is of concern that the visible images with low intensity at night were often contaminated by noise, and the corresponding fusion images may have been affected or even have their noise enhanced, such as in the fourth row in Figure 6. CVT and IFEVIP were less affected by noise, but their edges were blurred, and the details were not clear enough. STF was most disturbed by noise, especially in a dark background. In addition, the targets were not significant enough, which would affect the subsequent detection performance. VSMWLS and the proposed method were affected by noise to some extent, but the highlighted target areas and rich detail information made the targets still prominent. Therefore, compared with the other methods, the proposed method effectively highlighted the thermal target and retained the visible details of the target area, which is beneficial to pedestrian detection, especially in scenes with complex illumination.

To quantitatively compare the fusion performance of different methods, four evaluation metrics, including the multi-scale correlation coefficient (MCC) [31], spatial frequency (SF) [10], entropy (EN) [52], and average gradient (AG) [11], were used to qualitatively compare our method with other methods. The MCC evaluates the fusion method with the measurements of the correlation of the fused result and the input images at different scales. The SF assesses the details and textures of the fused result based on gradient distribution. EN evaluates the quantity of the information of the fused result. The AG reflects the definition of the image by gradient calculation. Larger values for these four metrics imply better fusion performance.

A quantitative comparison of the different fusion methods is shown in Table 1. It can be observed that the proposed method achieved the maximum MCC, SF, and AG in most cases, which means that the fusion results of this method had the highest degree of correlation between the fusion results and source images, the best clarity, and the most detailed information. Aside from that, our method achieved the maximum EN value in many cases, indicating that the fusion results contained a lot of information. It should be noted that the proposed method is usually unable to obtain high EN values with intricate visible backgrounds, such as the fusion results in the fifth and sixth row in Figure 6. Compared with the proposed method, other methods retained more background information in the visible image, but this would make the targets less significant and affect the subsequent detection task.

That aside, it is worth noting that although the KAIST dataset was relatively well aligned, about 10% of the image pairs were still misaligned [53], which affected the image fusion, as shown in Figure 7. As can be seen from the figure, for the fusion of misaligned visible infrared image pairs, all the methods were affected by the retention of unaligned image features to varying degrees. The target areas of the fusion results of CVT, IFEVIP, VSMWLS, and STF were "overlapped" because the unaligned intensity information was retained, which may have interfered with the subsequent detection of pedestrian targets. The proposed method mainly preserved the intensity features of the infrared images, so it was minimally affected by the misregistration of the source images. In summation, our proposed method possessed the best fusion performance, and its fusion results were conducive to subsequent pedestrian detection.

**Table 1.** Quantitative comparison of different fusion methods.

| Images | Metrics | CVT | IFEVIP | VSMWLS | STF | Proposed |
|---|---|---|---|---|---|---|
| IM1 | MCC | 1.1918 | 1.0927 | 1.1701 | 1.3349 | 1.3553 |
| | SF | 7.8644 | 10.671 | 10.0784 | 9.9651 | 12.2615 |
| | EN | 6.8309 | 7.1229 | 7.1906 | 7.2936 | 7.36 |
| | AG | 2.4887 | 3.4168 | 3.4674 | 3.7597 | 4.1013 |
| IM2 | MCC | 1.5983 | 1.5083 | 1.5617 | 1.6187 | 1.637 |
| | SF | 9.0672 | 12.5468 | 11.9432 | 11.1744 | 14.2152 |
| | EN | 7.5442 | 7.5929 | 7.6619 | 7.4994 | 7.6885 |
| | AG | 3.4018 | 4.5238 | 4.6586 | 4.5967 | 5.353 |
| IM3 | MCC | 1.4016 | 0.6532 | 1.4086 | 1.5853 | 1.6559 |
| | SF | 7.4361 | 12.4645 | 9.6137 | 16.4532 | 14.9262 |
| | EN | 6.5595 | 7.1805 | 7.0967 | 6.6205 | 6.8944 |
| | AG | 2.1096 | 3.424 | 3.0389 | 4.4684 | 4.105 |
| IM4 | MCC | 1.1097 | 1.0094 | 1.1994 | 1.0071 | 1.1443 |
| | SF | 5.8053 | 8.1269 | 7.747 | 10.3357 | 8.2588 |
| | EN | 6.7783 | 7.0253 | 6.9894 | 6.9619 | 7.123 |
| | AG | 2.4294 | 3.4705 | 3.5266 | 4.3216 | 3.7777 |
| IM5 | MCC | 0.9759 | 1.0465 | 1.0513 | 1.0216 | 1.0668 |
| | SF | 6.6447 | 10.0782 | 9.4477 | 7.8325 | 10.6976 |
| | EN | 7.0949 | 7.3614 | 7.314 | 6.7392 | 6.8128 |
| | AG | 2.3316 | 3.4456 | 3.5094 | 3.3504 | 3.9016 |
| IM6 | MCC | 1.4509 | 1.4192 | 1.4462 | 1.4294 | 1.4681 |
| | SF | 7.5171 | 10.9581 | 10.658 | 8.6367 | 12.7359 |
| | EN | 7.3604 | 7.2918 | 7.5574 | 7.0863 | 7.1386 |
| | AG | 2.7683 | 3.8174 | 4.1147 | 3.314 | 4.6592 |

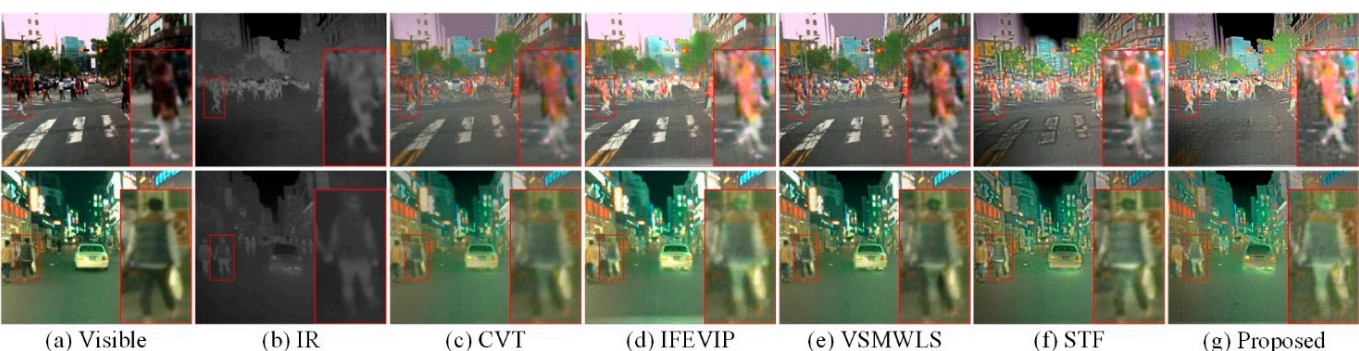

**Figure 7.** Fusion results of misaligned images. (**a**) shows the visible images; (**b**) shows the infrared images; (**c**–**g**) respectively show the fusion results based on CVT, IFEVIP, VSMWLS, STF and the proposed method.

### 3.3. Comparison of the Detection Results via Different Fusion Methods

In order to prove the detection performance of the method in this paper, the detection results based on the proposed method, CVT, IFEVIP, VSMWLS, STF, early fusion, and late fusion were evaluated, with the YOLOv3 detector via visible images as a benchmark. Following the evaluation protocol in [33], this paper used two indexes for the evaluation, namely the miss rate−false positive per image (MR−FPPI) curve and log−average miss rat. A lower MR−FPPI curve or lower log-average miss rate means better detection performance.

The detectors based on different fusion methods were tested on the KAIST test set, and the quantitative results are shown in Figure 8. It can be observed that for the daytime detection, the original detector based on visible images could obtain fairly good detection performance. CVT, IFEVIP, STF, and early fusion did not improve the detection performance and even worsened it. VSMWLS and late fusion improved the detection performance, but

the improvements were not obvious. Compared with the other five detectors, the detector based on the proposed method possessed the best performance of the daytime detection.

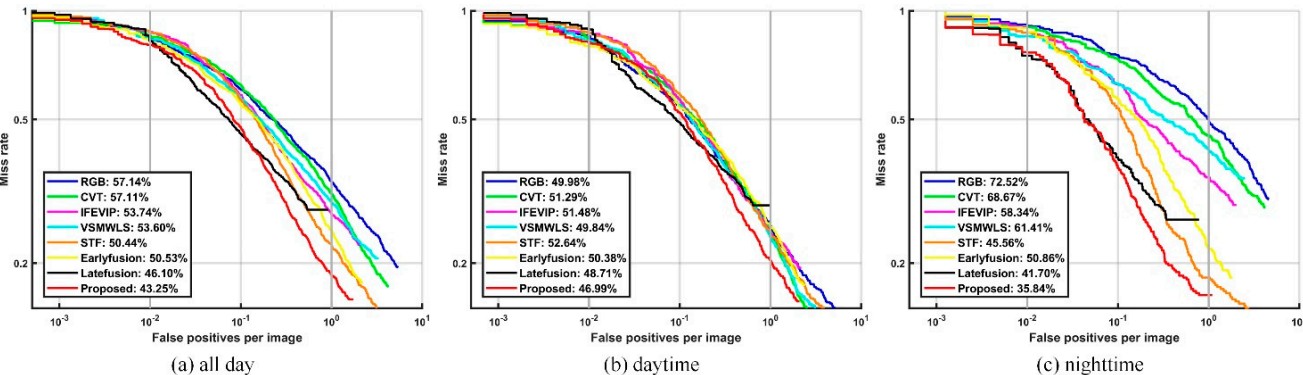

**Figure 8.** Quantitative comparison of the detection results via eight methods. (**a**) presents the results of all-day detection; (**b**) exhibits the results of daytime detection; (**c**) exhibits the results of nighttime detection.

For the nighttime detection, the performance of the original detector based on visible images was poor, and the log−average miss rate was as high as 72.52%. All the fusion methods could improve the performance of night detection to different degrees. The nighttime detection performance of CVT, IFEVIP, and VSMWLS was obviously inferior to the performance of STF, early fusion, and late fusion, but the proposed method still obtained the best performance of the nighttime detection.

In general, all the fusion methods could improve the robustness for all-time detection. The detection performance of the two fusion architectures was obviously better than that of CVT, IFEVIP, and VSMWLS, especially for the late fusion. The detection performance of STF was similar to that of early fusion. The daytime detection performance of early fusion was slightly better, while STF had better nighttime detection performance. However, the detection performance of our proposed method achieved the best detection performance, whether for daytime or nighttime detection.

Figure 9 demonstrates the detection results of some images in the test set. Here, the first row shows the ground truth, the second row shows the detection results based on the visible RGB images, and the other rows show the detection results of CVT, IFEVIP, VSMWLS, STF, early fusion, late fusion, and the proposed method in turn. Aside from that, the first three columns are the daytime detection results, and the last three columns are the nighttime detection results.

As can be observed, compared with the other methods, the proposed method was more effective at reducing false detections and missed detections, especially in scenes with poor illumination conditions and chaotic backgrounds. In backlight scenes, part of the visual field will be occupied by the background light, and some pedestrians will be "invisible" in the image, as shown in the RGB image in the fifth column of Figure 9. The other six fusion methods could not effectively distinguish the background from the targets, so the pedestrian under the backlight condition could not be detected. However, the proposed method could ignore the obvious background features and retain the significant features of the targets, meaning it could detect the pedestrians accurately even under the backlight condition.

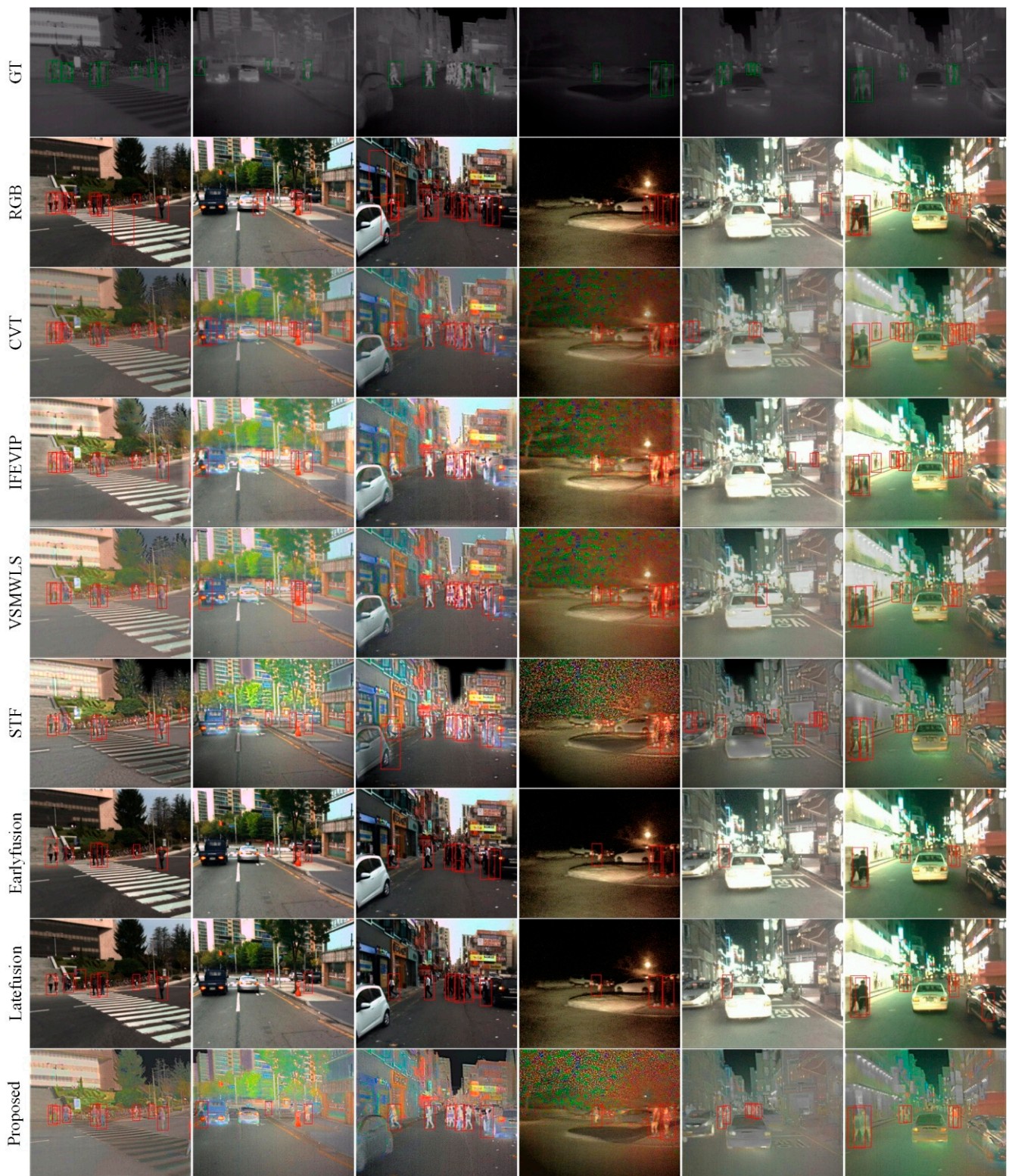

**Figure 9.** Qualitative comparation of the detection results via seven methods.

That aside, it is worth noting that, as shown in the fourth column in Figure 9, the noise of the visible image at night would be enhanced to varying degrees in the fusion results. STF was the most affected by noise because it retained a large amount of visible information, and the small-scale target detection was seriously disturbed by background noise. The fusion image of the proposed method was also affected by noise, but the

prominent target area and clear details helped this method still achieve better detection performance compared with that of other methods. In summation, compared with the other methods, the proposed method had superior detection performance and could effectively utilize the features of the infrared and visible images, thus improving the accuracy and robustness of pedestrian detection.

## 4. Conclusions

In order to explore a better multi-spectral pedestrian detection method, several detectors based on different fusion methods were studied and tested in this paper. A method for robust pedestrian detection based on multi-spectral image fusion and YOLOv3 was proposed. In this method, TV minimization based on structure transfer is adopted to combine infrared images and visible images, preserving the infrared intensity distribution and local appearance information. Then, the infrared detail enhancement is used to achieve fusion images with prominent targets and abundant details. Multi-spectral pedestrian detection is realized by combining the fusion images with YOLOv3. Aside from that, two fusion architectures based on YOLOv3 were designed and implemented which fused the features of different scales in different depths of the network. At last, the KAIST dataset was adopted to evaluate the detection performance of our proposed method, the other four fusion methods, and two fusion architectures. The results demonstrate that our proposed method effectively improved the robustness and accuracy of pedestrian detection and still had a good detection effect in challenging, complex scenes.

It should be noted that this paper assumed that there was a significant difference between the thermal radiation of the pedestrian targets and the background; that is, the targets in the infrared images were significant. Under this premise, the proposed method could make full use of the infrared intensity information and visible detail information to achieve accurate and robust all-time pedestrian detection. However, when the pedestrian targets are not prominent in infrared images, the detection performance of the proposed method will be affected. This problem is mainly limited by the characteristics of infrared imaging, and it may be solved by combining the information of other modalities. That aside, the computational efficiency of the method in this paper cannot meet the demand of real-time detection, and further research is needed for this problem.

**Author Contributions:** Conceptualization, X.C.; data curation, X.T.; formal analysis, X.C.; funding acquisition, L.L.; investigation, X.T.; methodology, X.C.; project administration, L.L.; resources, X.T.; software, X.C.; supervision, L.L.; validation, X.C.; visualization, X.T.; writing—original draft, X.C.; writing—review and editing, L.L. All authors have read and agreed to the published version of the manuscript.

**Funding:** This research was funded by the LLL Night Vision Technology Key Laboratory Fund (grant No. J20190102) and the Qing Lan Project of Jiangsu province in China (Grant No. 2017-AD41779).

**Data Availability Statement:** Publicly available datasets were analyzed in this study. This data can be found here: https://github.com/SoonminHwang/rgbt-ped-detection.

**Conflicts of Interest:** There were no known competing financial interests or personal relationships that that may have affected this work.

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
