# Peer review of "Robust Pedestrian Detection Based on Multi-Spectral Image Fusion and Convolutional Neural Networks"

_electronics, doi:10.3390/electronics11010001_

Round 1

Reviewer 1 Report

The authors have presented a method for robust pedestrian detection based on multi-spectral image fusion and YOLOV3.

.The papers provides  correct and informative figures and tables easily .The conclusion is logically supported by the obtained simulated results.The present work may be accepted without English correction ,sufficient for publicationHowever I suggest strictly to include in the references section the publication already submitted:

Sciuto, G. L., Napoli, C., Capizzi, G., & Shikler, R. (2019). Organic solar cells defects detection by means of an elliptical basis neural network and a new feature extraction technique. Optik, 194, 163038.In that way the writers can also compare with other methodologies their activity.

Reviewer 2 Report

The main innovation of this paper is not clear enough.  

Did you apply any method for improving the image to noise ratio? any filter

I suggest authors to proofread the paper.

please add more recent related work 

 what’s the theoretical basis behind the converting  RGB to HSI image

Complexity time should be discussed

How the proposed model deals with noisy images.

The results presented are very good, but can possibly be achieved with other state-of-the-art methods for image signal processing

Please discus the limitations of the proposed model

Reviewer 3 Report

  • Can you please elaborate a bit on the computational complexities of the algorithms and how much time would it for the proposed algorithm for detection?
  • How would the algorithm perform on noisy images such as noise from the camera sensor (blur, exposure, codec error, darkening, dirty lens), rain, shadow, or added synthetic noise? Can you kindly add some results on noisy images in the paper?  

Please edit the paper so that symbols and equations are in line with the text (highlighted in the attached pdf).

Round 2

Reviewer 2 Report

Authors have addressed my comments, the revised paper can be accepted for publication